# Assessment of Alkali–Silica Reactivity of Aggregates by Concrete Expansion Tests in Alkaline Solutions at 38 °C

**DOI:** 10.3390/ma13020288

**Published:** 2020-01-08

**Authors:** Irene Bavasso, Umberto Costa, Teresa Mangialardi, Antonio Evangelista Paolini

**Affiliations:** 1Department of Chemical Materials Environment Engineering, Faculty of Civil and Industrial Engineering, La Sapienza University, 00184 Roma, Italy; irene.bavasso@uniroma1.it (I.B.); antonioevangelista.paolini@uniroma1.it (A.E.P.); 2Technical Association of Italian Cement Manufacturers, 24122 Bergamo, Italy; u.costa@live.com

**Keywords:** alkali–silica reaction, concrete aggregates, concrete expansion tests, alkaline solution, threshold alkali level

## Abstract

A new accelerated concrete prism expansion test at 38 °C (accelerated CPT) is proposed for assessing the alkali-reactivity of concrete aggregates. In this test, concrete prisms with a standardized mix composition and different alkali contents are immersed in alkaline solutions with compositions simulating the pore liquid of hardened concretes. The concrete prism expansion test at 38 °C and RH > 95% (traditional CPT) was taken as a reference test, in order to define the appropriate expansion limit criterion for the proposed accelerated CPT. Three natural aggregates of known field performance and different alkali–silica reactivity were tested. The compositions of alkaline solutions were designed by assuming total dissolution of cement alkalis and taking a ratio between the mass fractions of effective water consumed by cement hydration and of alkalis uptaken by cement hydrates equal to unity. This simplified approach was found in an acceptable agreement with literature empirical equations correlating pore solution alkalinity of hardened Portland cement mixes with total alkali content of cement. Elaboration of expansion data through both pass-fail and threshold alkali level (TAL)-evaluation approaches indicated that, for the accelerated CPT, an expansion limit criterion of 0.04% after 120 days of testing in alkaline solutions is appropriate to evaluate the aggregate alkali reactivity congruently with the traditional CPT. Use of the proposed test method in place of the traditional CPT would reduce the test duration from 365 to 120 days.

## 1. Introduction

Alkali–silica reaction (ASR) is one of the most frequent causes of concrete deterioration [1,2]. ASR is a slowly expansive reaction between certain forms of alkali-reactive silica (opaline silica, flint, cryptocrystalline quartz) and/or certain silicate minerals present in concrete aggregates and the hydroxyl ions in concrete pore solution, mainly associated with sodium and potassium ions. This reaction leads to the formation of an alkali–silicate gel, which absorbs water and swells, causing internal expansive pressure [3,4,5]. In order to prevent deleterious ASR development in new concrete structures, it is in advance necessary to accurately characterize concrete aggregates for their alkali-expansivity and, if required, to use appropriate mitigation measures in the design of durable concrete mixes.

At present, many test methods are available in the literature for assessing the alkali-reactivity of concrete aggregates [2,6]. Some test methods are based on the direct analysis of aggregates, such as the chemical methods and the petrographic examination. Other test methods are based on the expansive behaviour of cement–aggregate combinations, such as the mortar bar expansion test or concrete prism expansion test [2,6].

Among the various test methods proposed, the concrete prism expansion test at 38 °C and RH > 95%, also referred to as traditional CPT in this study, is considered the most reliable to assess both the alkali-reactivity of aggregates [2,6,7] and the efficacy of supplementary cementing materials (SCMs) in preventing or minimizing the development of deleterious ASR expansion [2]. This test has been standardized in many countries or proposed by scientific organizations, such as RILEM.

The most widely used versions of the traditional CPT are those published by CSA (CSA A23.2-14A) [8], ASTM (ASTM C 1293) [9], AFNOR (NF P18-594) [10], Standards Australia (AS 1141.60.2) [11] for unwrapped concrete prisms, or by RILEM (RILEM AAR-3) [6] for wrapped concrete prisms. In Italy, the traditional CPT has been standardized as UNI 11604 in 2015 [12].

This test is often taken as a reference test for calibrating other ultra-accelerated concrete prism test methods. However, the long duration of the traditional CPT (eight months or one year for aggregate reactivity assessment, two years for SCM efficacy assessment) represents a considerable drawback and renders this test impractical for use in quality control testing and evaluation of concrete aggregates and SCMs.

In order to accelerate the traditional CPT, two distinct modifications have been considered: Increasing the test temperature without altering the moisture conditions or providing a continuous supply of alkalis by immersion of concrete prisms in alkaline solutions without altering the test temperature.

These two approaches have inspired development of the concrete prism expansion test at 60 °C and RH > 95%, proposed by RILEM as the AAR-4.1 test method [6] and the concrete prism expansion test in a 1M NaOH solution at 38 °C, proposed by Touma et al. [13] as the modified ASTM C 1293 testing procedure.

However, published work [14,15,16,17,18] has shown that an increase in the test temperature from 38 to 60 °C leads to reduced expansion of concrete prisms. This reduction is ascribed to several factors including increased drying, reduced concentration of OH^−^ ions in the pore solution of concrete related to both ettringite decomposition at 60 °C and increased alkali leaching, and higher exudative properties of ASR gels (related to lower viscosity at 60 °C). The reduced expansion of concrete prisms could in turn lead to false-negative test results when aggregates with relatively low alkali-expansivity are tested using the accelerated CPT at 60 °C. For such aggregates, higher concentrations of OH^−^ ions in the pore solution of concrete are needed to develop deleterious ASR expansion (higher OH^−^ ion thresholds). Therefore, further research is needed to overcome the drawbacks of this test method. In this context, a reduction of the test temperature from 60 to 50 °C should also be considered [19,20].

On the other hand, the concrete prism expansion test in 1M NaOH solution at 38 °C may only be regarded as a pass-fail test and, therefore, it is not suitable for determining the threshold alkali level (TAL) of concrete aggregates, an important alkali-reactivity parameter that is specific to the aggregate tested [21,22]. The TAL parameter is defined as the minimum alkali content of the standardized concrete mix containing the test aggregate, expressed as kg Na_2_Oeq/m^3^, above which deleterious expansion of concrete will occur. The higher the TAL value, the lower the alkali-reactivity of the aggregate tested. The knowledge of the TAL parameter is very useful in designing durable concrete mixes incorporating ASR-susceptible aggregates [22,23].

The determination of the TAL parameter requires the execution of expansion tests on concrete mixes made with the test aggregate and different alkali contents. At this purpose, the test procedure of the traditional CPT may be appositely modified to convert this test from a pass-fail to TAL-evaluation test [6,12].

In this study, a new concrete prism expansion test in alkaline solutions at 38 °C is proposed in order to achieve a complete and rapid characterization of the alkali-reactivity of concrete aggregates. In the proposed test method, also referred to as accelerated CPT, the 1M NaOH solution is replaced with an alkaline (NaOH + KOH) solution simulating the pore liquid of the concrete prisms having a specific alkali content. It follows that, in the case of the aggregate TAL evaluation, alkaline solutions with different chemical compositions corresponding to the different alkali contents of the concrete mixes are used, while only one alkaline solution with an established chemical composition is used for the pass-fail alkali-reactivity assessment.

Three natural aggregates known for their different alkali-expansivity are tested using both the accelerated and traditional CPT with the aim of defining the appropriate expansion limit criterion for the proposed test method. At this purpose, alkali leaching from concrete prisms subjected to traditional CPT and OH^−^ ion concentration changes in the alkaline solutions during the accelerated CPT are also evaluated.

## 2. Materials and Methods

### 2.1. Cement and Aggregates

A low-alkali Portland cement (CEM I 42.5N) conforming to EN 197-1 [24] and three natural aggregates coming from Italian quarries were used to prepare concrete mixes with different ASR expansivity levels. Table 1 gives the chemical and mineralogical (Bogue method) composition of the cement.

The three aggregates tested were known to significantly differ for their ASR expansivity in the field and were designated as A1, A2, and A3. Aggregate A1 was known to be alkali-reactive, while aggregates A2 and A3 were known as not alkali-reactive. The choice of two not reactive and one reactive aggregates was made in order to better highlight eventual false-positive test results when comparing the diagnoses obtained with the accelerated and traditional CPT (expansion results above the limit only with accelerated CPT).

Each aggregate was available in both coarse and fine grain sizes.

Petrographic examination [6] showed that aggregate A1 consisted of fragments of medium-to-fine grained sedimentary carbonatic rocks (fossil micrite and sparite), flint in sub-millimeter clasts and chalcedony within the carbonate rocks pores. Aggregate A2 consisted of fragments of sedimentary carbonatic rocks and sandstones, altered feldspars, monocrystalline quartz, with chalcedony (fibrous microcrystalline silica) and flint (porous microcrystalline silica) as the main alkali-reactive phases. Aggregate A3 consisted of metamorphic (quartzitic and metadioritic) rocks and igneous (rhyolite/dacite) rocks.

On the basis of the petrographic examination and taking into account the alkali-reactivity classification proposed by RILEM AAR-0 [6], aggregates A1 and A2 were classed as Class III-S (very likely to be alkali–silica reactive), aggregate A3 was classified as Class I (very unlikely to be alkali–silica reactive). As discussed later (Section 4.3.1 and Section 4.3.2), the results of the traditional CPT (pass-fail and TAL-evaluation approaches) confirmed the different field performances of the three aggregates (aggregate A1 as alkali-reactive, aggregates A2 and A3 as not alkali-reactive).

### 2.2. Concrete Mixes and Expansion Test Procedures

Concrete mixes were made using Portland cement, the test aggregate in both fine (0–4 mm) and coarse (4–22.5 mm) size gradations and deionized water as mixing water. The concrete mix proportions were those specified in the RILEM AAR-3 and UNI 11604 test methods [6,12]. Appropriate additions of NaOH pellets to mixing water were also made to increase the level of alkali content, Lac, of the concrete mixes to the desired values (alkali-boosted mixes). Table 2 summarizes the compositions of the concrete mixes investigated. The Lac values reported in this table correspond to the initial alkali contents of the unboosted or alkali-boosted concrete mixes (initial Lac).

The unboosted concrete mix was characterized by a Lac value of 2.73 kg Na_2_Oeq/m^3^ that was calculated on the basis of cement content (440 kg/m^3^) and its total alkali content (0.62% Na_2_Oeq), and by neglecting the amount of alkalis released from the test aggregate. This is because the release of alkalis from aggregates is much slower as compared to Portland cement. Moreover, at present, there are no validated test methods to quantify such a release.

Depending on the alkali-reactivity of the aggregate tested, the Lac value was varied over the range from 2.73 to 5.5 kg Na_2_Oeq/m^3^ for the concrete mixes made with aggregate A1, over the range from 2.73 to 9.5 kg Na_2_Oeq/m^3^ or from 5.5 to 9.5 kg Na_2_Oeq/m^3^ for the concrete mixes made with aggregates A2 or A3, respectively.

From each concrete mix, six prismatic specimens (70 × 70 × 260 mm in size) were cast and preliminarily cured for four days (one day within the moulds) in a room kept at 20 ± 2 °C and RH > 95%. Afterwards, the prisms were measured for their initial length (using a length comparator with a sensitivity of 0.001 mm), and then placed inside airtight, stainless steel, cylindrical containers (internal diameter = 13.5 cm; internal height = 31.5 cm) as designed according to UNI 11604 [12]. One unwrapped concrete prism was placed in each container and stored vertically on a perforated rack at about 30 mm from the container bottom.

In the case of the concrete prisms to be subjected to the traditional CPT (three specimens), a 200-mL volume of deionized water was added in the container (water reservoir) and no wick of absorbent material in contact with the water reservoir was placed around the inner wall of the container.

In the case of the remaining three concrete prisms to be subjected to the accelerated CPT, a 2.8-liter volume of alkaline solution (NaOH + KOH solution) with appropriate chemical composition was added to each container, so that each concrete prism was completely submerged.

The chemical composition of the alkaline solutions was designed on the basis of the cement alkali content (Table 1) and concrete mix composition (Table 2), taking in mind that the pore liquid of concrete essentially consists of a solution of NaOH + KOH with sulphate, chloride and, even more, calcium and magnesium as minor ionic species [25]. In the preparation of the test solutions, the minor ionic species were not considered and the following relationship between the initial Lac (kg Na_2_Oeq/m^3^) of concrete mixes and the corresponding OH^−^ ion concentration (moles/L) in alkaline solutions was used:(1)$$\left[{\mathrm{OH}}^{-}\right]=\;\frac{\mathrm{Lac}}{31\cdot V_{\mathrm{pl}}}$$
where 31 is the equivalent weight of Na_2_O and V_pl_ is the pore liquid volume (m^3^) per unit volume (m^3^) of concrete. The V_pl_ value was taken equal to the volume of effective water used per unit volume of concrete mix (0.22 m^3^/m^3^ of fresh concrete) (Table 2).

Reagent-grade NaOH and KOH pellets and deionized water were used to prepare the designed alkaline solutions, whose compositions are given in Table 3, along with the respective concrete alkali contents (initial Lac).

After closure, all the containers were placed in a laboratory oven equipped with air ventilation and operated at a controlled temperature of 38 ± 2 °C.

At established testing times, concrete prism length measurements were made after cooling the closed containers for about 16 h in a room kept at 20 ± 2 °C. The percentage expansion of each concrete prism was determined from the length change at each testing time, and the average value of expansion was then calculated on triplicate specimens. For average expansions of less than 0.02%, the standard deviation was lower than 0.0035%. For average expansions of more than 0.02%, the coefficient of variation did not exceed 15%.

### 2.3. Monitoring of OH^−^ Ion Concentration in Alkaline Solutions during the Accelerated CPT

The OH^−^ ion concentration in the alkaline solutions (NaOH + KOH solutions) during the accelerated CPT was evaluated at different immersion times of concrete prisms. At this purpose, aliquots (10 mL) of each alkaline solution were subjected to volumetric titration with standardized nitric acid solution using phenolphthalein and/or methyl-orange as indicators. These determinations were made to estimate the suitability of the test apparatus in attenuating the changes in the OH^−^ ion concentration of the external alkaline solutions during the immersion test.

### 2.4. Alkali Leaching in the Traditional CPT

The alkali (Na_2_O, K_2_O) leaching from each concrete prism subjected to the traditional CPT was evaluated after one year of testing by measuring the concentrations of alkali metal ions (Na^+^ and K^+^) in the water reservoir of each container through the use of a flame atomic absorption spectrophotometer (AAS) [26]. The amount of alkalis leached was calculated taking into account the volume of water reservoir measured in each container after one year of testing (170–175 mL).

## 3. Results

### 3.1. Accelerated CPT Method

#### 3.1.1. Changes in OH^−^ Ion Concentration of Alkaline Solutions during the Tests

During the immersion tests, the diffusion of alkalis in the form of NaOH and KOH from the external solution into the pore liquid of the concrete prisms is able to balance the alkali uptake by cement hydration products and the alkali consumption by ASR, so that the alkali–silica reaction will be sustained.

The diffusion rate of alkalis is commonly much higher than the rate of ASR expansion (different time scales). Therefore, the OH^−^ ion concentration in the pore liquid of the concrete prisms may be assumed to be approximately equal to the OH^−^ ion concentration in the external alkaline solution at the immersion time considered.

Obviously, the diffusion of alkalis produces a progressive reduction of OH^−^ ion concentration in the external solution during the accelerated CPT. This reduction depends on the amount of alkalis consumed by ASR and uptaken by cement hydrates and, to a much greater extent, on the ratio between the external alkaline solution volume (V_as_) and the initial pore liquid volume (V_pl_) of the concrete prism used in the test. The initial pore liquid volume is, in turn, dependent on the water/cement ratio of the concrete mix and the volume of the prism tested. During the immersion test, cement hydration reduces progressively the pore liquid volume of concrete prisms so that, to calculate the actual Lac value of concrete by using Equation (1), the residual pore liquid volume should be known.

Figure 1 shows the OH^−^ ion concentrations of the external alkaline solutions plotted as a function of the immersion time of concrete prisms made with different initial alkali contents, over the whole range of initial Lac values investigated (2.73–9.5 kg Na_2_Oeq/m^3^). In particular, the data reported in this figure are relevant to concretes with aggregate A1 (Lac values from 2.73 to 5.5 kg Na_2_Oeq/m^3^), aggregate A2 (Lac values of 5.5 and 7.5 kg Na_2_Oeq/m^3^), and aggregate A3 (Lac values of 6.4 and 9.5 kg Na_2_Oeq/m^3^).

As expected, the OH^−^ ion concentrations in the external alkaline solutions decreased with increasing immersion time of concrete prisms but these reductions remained very low even after long immersion times. At the ultimate testing time adopted in this study (420 days), the percentage reduction of OH^−^ ion concentration of external solutions were 4.5–13%. Lower percentage reductions of OH^−^ ion concentrations (2–9.5%) were found after 180 days of immersion.

These data indicated that the test conditions adopted in this study for the accelerated CPT, in particular, the alkaline solution/initial pore liquid volume ratio (V_as_/V_pl_ = 10), were suitable to strongly attenuate the OH^−^ ion concentration changes of the external solution during the immersion test. Changes in OH^−^ ion concentration of the alkaline solutions could be further attenuated by using a testing apparatus with higher V_as_/V_pl_ values.

Therefore, the accelerated CPT could roughly be regarded as a concrete prism expansion test at an approximately constant concentration of OH^−^ ions in the pore liquid of concrete, with this concentration being related to the initial Lac of the concrete mix. This is because the OH^−^ ion concentration in the pore liquid is approximately equal to the OH^−^ ion concentration of the external alkaline solution during the test and the latter may be approximately taken equal to its initial concentration (Figure 1). The initial OH^−^ ion concentration of the alkaline solution is, in turn, related to the initial Lac of concrete mix by Equation (1).

#### 3.1.2. Expansions of Concrete Prisms

Figure 2a–c show the expansion data obtained using the accelerated CPT on the concrete mixes made with aggregate A1 (Figure 2a), aggregate A2 (Figure 2b), or aggregate A3 (Figure 2c).

These figures depict the effects of increasing the immersion time or the concrete alkali content (initial Lac) on the expansion of concrete prisms.

As expected, for the concrete prisms made with a given aggregate, expansion was found to increase with increasing both prism immersion time and concrete alkali content. However, irrespective of the aggregate type and concrete alkali content considered, no expansion curve levelled off at least up to the ultimate testing time (420 days). This was particularly evident for the concrete prisms made with aggregate A1 (Figure 2a) that was the most alkali-reactive.

At 420 days of testing, the expansion level (0.079%) of the concrete prisms made with aggregate A3 and Lac = 9.5 kg Na_2_Oeq/m^3^ (Figure 2c) was still much lower than the expansion (0.40%) measured for the concrete prisms made with aggregate A1 and Lac = 5.5 kg Na_2_Oeq/m^3^ (Figure 2a) thus evidencing the very low alkali-reactivity of the former aggregate.

Concretes with aggregate A2 showed expansion levels of less than 0.04% after 420 days of testing only for the alkali-unboosted mix (Figure 2b).

When the expansions at 420 days (ultimate expansions) of concrete mixes made with aggregates A1 and A2 at Lac = 5.5 kg Na_2_Oeq/m^3^ (Figure 2a,b) were related to the respective alkali consumptions in terms of OH^−^ ions evaluated from the immersion data of Figure 1, an inverse relationship between alkali consumption and aggregate expansivity was found. In particular, the OH^−^ ion consumptions per unit volume of concrete prism were equal to 80.8 and 197.8 mmoles/L of concrete against ultimate expansions of 0.40% and 0.156% for aggregates A1 and A2, respectively. This observation is congruent with the aggregate alkali-reactivity assessment based on the TAL parameter (the higher the aggregate reactivity, the lower the TAL value).

### 3.2. Traditional CPT Method

#### 3.2.1. Expansions of Concrete Prisms

Figure 3a–c show the time development of the expansion of the concrete mixes of Table 2 subjected to the traditional CPT. For comparison purpose with the accelerated CPT, testing time was prolonged up to 420 days (ultimate testing time), even if the alkali reactivity assessment with the traditional CPT is often based on a testing time of 365 days [6,8,9,12].

The results in Figure 3 confirmed the different alkali-reactivity of the three aggregates tested: Aggregate A1 (Figure 3a) was the most expansive and aggregate A3 (Figure 3c) the least expansive. At a fixed alkali content (Lac = 5.5 kg Na_2_Oeq/m^3^), the expansion levels of concrete prisms made with aggregate A1 were found to be much higher than those observed by concrete prisms made with aggregates A2 or A3 over the whole range of testing times examined.

At 420 days of testing, the expansion level (about 0.14%) of concrete prisms with aggregate A1 and alkali content of 5.5 kg Na_2_Oeq/m^3^ was still higher (about two-fold) than those measured for the concretes made with aggregates A2 or A3 at the highest alkali content investigated (Lac = 9.5 kg Na_2_Oeq/m^3^).

Similarly to what happened in the accelerated CPT, no expansion curve levelled off at least up to the ultimate testing time, except for the concrete prisms made with aggregate A2 (Figure 3b).

Irrespective of the aggregate tested, the ultimate expansion levels always increased with increasing concrete alkali content.

Comparison between the results of Figure 2 and Figure 3 indicated that, at a fixed concrete alkali content, the ultimate expansions of concrete prisms subjected to accelerated CPT were always much higher than those obtained using the traditional CPT. At the concrete alkali content of 5.5 kg Na_2_Oeq/m^3^, the ratios between the ultimate expansions obtained with the accelerated and traditional CPT were about 2.8, 11, and 24 for the concrete mixes made with aggregates A1, A2, and A3, respectively. As expected, the ratio between the ultimate expansions from the two test methods decreased with increasing aggregate reactivity.

#### 3.2.2. Alkali Leaching

Differently from what occurs in the accelerated CPT, alkali consumption by ASR, alkali uptake by cement hydrates, and alkali leaching from concrete prisms are all responsible for a progressive reduction of the OH^−^ ion concentration in the pore liquid of concrete during the traditional CPT.

Table 4 gives the amounts of alkaline metal ions, Na^+^ and K^+^, leached out of the concrete prisms after one year of testing with the traditional CPT (kg metal ion leached/m^3^ concrete). The total amounts of alkaline metal ions (Na^+^ + K^+^) leached are also reported in terms of alkalis (kg Na_2_Oeq/m^3^ concrete), along with the corresponding percentage Na_2_Oeq releases.

The leaching data relevant to aggregates A1 and A3 are the same as those published in our previous paper [27] dealing with the alkali leaching from the Portland cement concrete subjected to the traditional CPT.

For concrete mixes made with a given aggregate, increasing concrete alkali content (initial Lac) resulted in a little increase of alkali leaching. At a fixed concrete alkali content (Lac = 5.5 kg Na_2_Oeq/m^3^), the highest percentage alkali release (10.4%) was recorded for the concrete mix made with the most expansive aggregate (aggregate A1) and this was ascribed to crack formation [27] and ASR gel exudation [18]. However, in virtue of the appropriate design of the test apparatus, in particular the low ratio between the net container volume, V_nc_, and the concrete prism volume, V_c_ (V_nc_/V_c_ = 2.3), the alkali leaching was always found to be very low (percentage Na_2_Oeq releases of 4.9–11%), as compared to alkali leaching (35–37% Na_2_Oeq releases) reported by Thomas et al. [7] using the CSA A 23.2-14A test procedure [8] and by Lindgård et al. [28] using the ASTM C1293 test procedure [9].

The reduced alkali leaching associated to the use of containers with a low V_nc_/V_c_ ratio will improve the reliability of the traditional CPT as a reference test method.

## 4. Discussion

### 4.1. Verification of Consistency of the Experimental Approach in the Accelerated CPT Development

In the proposed accelerated CPT, different alkaline solutions were used to simulate the pore liquid compositions of concrete mixes made with different alkali contents (initial Lac).

A simplified approach based on Equation (1) (standardized concrete mix composition and Lac as unique variable) was used to prepare such alkaline solutions.

In order to verify the consistency of this simplified approach, the OH^−^ ion concentrations of the external alkaline solutions used in the immersion tests (Table 3) were compared with those calculated for concrete pore solution by using the empirical equation proposed by Helmuth [29]:(2)$$\left[{\mathrm{OH}}^{-}\right]=\frac{0.339{\;\mathrm{Na}}_{2}\mathrm{Oeq}\%}{w/c}+0.022$$
or the empirical equation proposed by Thomas et al. [7]:(3)$$\left[{\mathrm{OH}}^{-}\right]=0.70{\;\mathrm{Na}}_{2}\mathrm{Oeq}\%$$

In these two equations, [OH^−^] is the hydroxyl ion concentration (moles/L) in the pore solution, Na_2_Oeq% is the total alkali content of Portland cement used in the cementitious mix and w/c is the water/cement weight ratio of the considered mix.

Equation (2) was derived by Helmuth [29] through a multiple linear regression of test results obtained by various researchers on pore solutions expressed from 31 different mature cement pastes or mortars made with Portland cements having total alkali contents in the range from 0.2% to 1.4% as Na_2_Oeq. The standard error of the OH^−^ ion concentrations was reported to be ± 0.06 moles/L.

Thomas et al. [7] developed Equation (3) based on a review of pore solution data from Nixon and Page [30]. In Equation (3) the w/c ratio is equal to 0.50 and it is included in the proportionality constant (0.70).

In the present study, the w/c ratio of the concrete mixes was always equal to 0.50, while the concrete alkali content, Lac, was varied over the range from 2.73 to 9.5 kg Na_2_Oeq/m^3^ by adding appropriate amounts of NaOH pellets to mixing water.

To apply Equation (2) or Equation (3) to the concrete mixes of Table 2, the Lac values were converted into the corresponding Na_2_Oeq% values of alkali-boosted Portland cement through the following equation:(4)$${\mathrm{Na}}_{2}\mathrm{Oeq}\%=100\;\frac{\mathrm{Lac}}{c}$$
where c is the content of Portland cement per cubic meter of concrete mix (440 kg/m^3^ in this study).

Thus, the Na_2_Oeq% values varied over the range from 0.62% (alkali unboosted Portland cement; Table 1) to 2.16% (alkali-boosted Portland cement; Lac = 9.5 kg Na_2_Oeq/m^3^).

Using the Lac values of Table 3, Equation (4), and Equation (2) (with w/c = 0.50) or Equation (3), the OH^−^ ion concentrations of the concrete pore solutions were calculated.

Figure 4 compares the OH^−^ ion concentrations in the external alkaline solutions calculated by Equation (1) (Table 3) with those calculated for the concrete pore solution over the whole range of the Lac values investigated.

In this figure, the two dotted lines indicate the uncertainty interval (± 0.06 moles/L) for the OH^−^ ion concentration calculated with Equation (2) up to the Lac value of 6.2 kg Na_2_Oeq/m^3^, corresponding to the maximum total alkali content of Portland cements (1.4% Na_2_Oeq) investigated by Helmuth [29].

As evidenced by the results of Figure 4, the designed OH^−^ ion concentration values for the external alkaline solutions (Table 3; Equation (1)) fall within the uncertainty interval of Equation (2) relevant to the OH^−^ ion concentration of pore liquid. Even if the designed OH^−^ ion concentrations for the external solutions are systematically lower than those calculated for the pore liquid by Equation (3), the percentage differences are relatively low (7.5–9.7%). Therefore, the simplified approach based on Equation (1) for designing the composition of the external alkaline solutions simulating the composition of the pore liquid may be considered acceptable.

The results of Figure 4 coupled with those of Figure 1 suggested that, in the TAL-evaluation approach (Section 4.3.2.), it is possible to directly relate the expansion data obtained from the accelerated CPT to the initial Lac of the concrete mixes investigated.

### 4.2. Factors Affecting Pore Liquid Alkalinity

In order to understand the above systematic difference between the OH^−^ ion concentration of external solutions and concrete pore solutions, the factors affecting pore liquid alkalinity of concrete were considered and a modified form of Equation (1) was proposed.

Indeed, Equation (1) represents a simplified approach to calculating the OH^−^ ion concentration in the pore solution of Portland cement concrete. This equation assumes the total and prompt dissolution of cement alkalis and does not take into account: (1) The alkali uptake by the cement hydration products, especially calcium silicate hydrate (C–S–H gel) and, to a minor extent, calcium alumina–silicate hydrate (C–A–S–H) and monosulphate phase (AF_m_) [31,32,33,34,35] and (2) the consumption of effective water by cement hydration.

Thermodynamic modelling [36], as well as semi-empirical models [34] based on the interaction between C–S–H gel and pore liquid have been proposed for calculating the alkali uptake by C–S–H gel. The amount of water combined during Portland cement hydration may be evaluated using well-known relationships concerning capillary and total porosity of Portland cement pastes [25].

In the case of ASR investigations on Portland cement concrete, the distinction between the dissolution rates of the various chemical species contributing to the total alkali content of cement is not very significant for evaluating the effect of pore liquid composition on the expansion of concrete specimens. This is because Portland cement hydration and ASR expansion develop over different time scales. Portland cements may exhibit hydration degrees of 70–80% after 70–100 h of curing at temperatures ranging from 25 to 40 °C [37,38]. Conversely, depending on the aggregate reactivity and concrete mix composition, ASR develops after laboratory testing times ranging from a few weeks to several months [2]. For example, a latency time of 120 days has been reported for ASR development at 38 °C and RH > 95% in concrete containing siliceous limestone aggregate (w/c = 0.50) [39]. Moreover, concrete expansion is often in progress even after one year of testing under the above exposure conditions [2,6]. The appearance of alkalis (Na_2_O, K_2_O) in the hydrated forms (NaOH and KOH) in the pore liquid of concrete is always related to the hydration rate of Portland cement constituents. In particular, alkalis incorporated within some constituents such as C_3_A and C_2_S are released into the pore liquid only when hydration of these constituents occurs. On the other hand, the rapid dissolution of highly water-soluble chemical species such as Na_2_SO_4_ and K_2_SO_4_ does not contribute to the concentration of NaOH and KOH in the pore liquid until portlandite is formed from hydration of calcium silicate constituents, C_3_S and C_2_S and, successively, calcium trisulpho-aluminate hydrate (ettringite) precipitates.

The distinction between the various forms of alkalis (water-soluble and total alkalis) contributing to the pore solution alkalinity may only be useful to differentiate the OH^−^ ion concentrations in the pore liquid during the early and mature phases of Portland cement hydration, especially in the case of Portland cements very rich in water-soluble alkalis [35].

Based on the above considerations, Equation (1) was modified by combining this equation (with V_pl_ = w) with Equation (4) and by including two parameters, β_a_ and β_w_, which are respectively related to the amounts of alkalis and of the effective water available in the pore solution at the cement hydration time considered. In particular, β_a_ is the mass fraction of total alkali content of Portland cement (Na_2_Oeq%) available in the concrete pore solution and β_w_ is the mass fraction of effective water in the hardened concrete at the cement hydration time considered.

Equation (1) in the modified form is as follows:(5)$$\left[{\mathrm{OH}}^{-}\right]=\;\frac{1000\;\cdot c\;\cdot \beta_{a}\cdot {\mathrm{Na}}_{2}\mathrm{Oeq}\%}{31\;\cdot 100\cdot \beta_{w}\cdot w}=0.323\;\frac{c\;\cdot \beta_{a}\;\cdot {\mathrm{Na}}_{2}\mathrm{Oeq}\%}{\beta_{w}\;\cdot w}$$
where 0.323 is the conversion factor from kg Na_2_Oeq/m^3^ to moles/L OH^−^ and the other terms are as previously defined. Equation (5), for w/c = 0.50, reduces to Equation (1) through Equation (4) assuming hypothetically β_a_/β_w_ = 1.

It must also be considered that the alkali uptake by C–S–H gel from the pore solution is directly proportional to the amount of gel formed and to its Ca/Si mole ratio [31]. Furthermore, the amount of effective water combined in Portland cement hydrates is directly proportional to the amount of hydrates formed (mainly, C–S–H) [25]. Thus, for saturated, mature Portland cement concretes, the β_a_/β_w_ ratio should not significantly differ from the unity, especially if the alkalis in the water consumed during hydration are embedded in the gel.

As discussed in the preceding section, there is a little but systematic difference between the OH^−^ ion concentrations of the external alkaline solutions ([OH^−^]_ext_) designed by Equation (1) and the OH^−^ ion concentrations in the pore solutions calculated by Equation (3) ([OH^−^]_p_). The [OH^−^]_p_/[OH^−^]_ext_ ratio varies from 1.075 to 1.097 and on average is equal to 1.084. Therefore, a correction factor of 1.084 should be applied to Equation (1) for better designing the external alkaline solutions to be used in the proposed accelerated CPT. However, further investigation including pore liquid extraction and analysis on concrete specimens with different Lac values is needed to investigate about the relationship between the [OH^−^]_p_/[OH^−^]_ext_ and β_a_/β_w_ ratios under the specific conditions of the test method.

### 4.3. Elaboration of Expansion Data from Traditional and Accelerated CPT Methods

The expansion results from the accelerated CPT (Figure 2a–c) were compared with those obtained from the traditional CPT (Figure 3a–c), the latter test being taken as a reference test, in order to define the appropriate expansion limit criterion for the proposed accelerated CPT.

Both the pass-fail and the TAL-evaluation approaches were used for this elaboration of the test results.

#### 4.3.1. Pass-Fail Approach

According to UNI 11604 [12] and other standardized versions of the traditional CPT [8,9], an expansion limit of 0.04% after one year of testing was taken as judgement criterion for assessing the alkali-reactivity of the three aggregates investigated with this test method.

In the pass-fail approach, an alkali content of 5.5 kg Na_2_Oeq/m^3^ is required for the concrete prisms to be subjected to the traditional CPT according to RILEM AAR-3 [6] and UNI 11604 [12].

As evidenced by the expansion data of Figure 3 relevant to the concrete mixes with Lac = 5.5 kg Na_2_Oeq/m^3^, one year expansion of the concrete prisms incorporating aggregate A2 (Figure 3b) or aggregate A3 (Figure 3c) resulted to be lower than 0.04% while one year expansion of greater than 0.04% was recorded for the concrete mix made with aggregate A1 (Figure 3a).

Thus, according to the traditional CPT (pass-fail approach), aggregate A1 was classed as alkali-reactive, aggregates A2 and A3 as not alkali-reactive. This classification was in agreement with the known field performances of the three aggregates and, for aggregates A1 and A3, even with their petrographic classification.

As reported in Table 3, an alkaline solution with 25 g/L NaOH and 10.13 g/L KOH (corresponding to OH^−^ ion concentration of 0.81 moles/L) is required to simulate the initial pore liquid composition of the concrete prisms made with Lac = 5.5 kg Na_2_Oeq/m^3^.

Based on the test results obtained in the present study (Figure 2a–c), and the test results reported by Touma et al. [13] for the accelerated CPT using 1M NaOH solution at 38 °C, four different expansion limit criteria were taken into consideration in order to reach a judgement of reactivity for the tested aggregates congruent with that of the traditional CPT.

These criteria were as follows: 0.04% after 90 days, 0.04% after 120 days, 0.04% after 150 days, or 0.04% after 180 days of testing. The last criterion was the same as the one proposed by Touma et al. [13] for correlating the expansion data obtained from the concrete prism expansion test in 1M NaOH solution at 38 °C and the traditional CPT in the ASTM version [9].

Figure 5 shows the expansions of the concrete prisms immersed in alkaline (NaOH + KOH) solution with OH^−^ ion concentration of 0.81 moles/L after 90, 120, 150, or 180 days of testing with the accelerated CPT, plotted against the corresponding one year expansion of the concrete prisms (Lac = 5.5 kg Na_2_Oeq/m^3^) obtained from the traditional CPT.

The horizontal and vertical dotted lines corresponding to an expansion level of 0.04% for the two test methods are also drawn. Thus, this figure is divided into four zones. Zones I and III are both congruity zones: Zone I indicates not alkali-reactivity and zone III alkali-reactivity. Zone II and IV are both incongruity zones.

It can be seen from Figure 5 that, in the case of concrete prisms made with aggregate A2, the use of the expansion limit criterion of 0.04% after 180 days of testing was not suitable to correlate the test results of the accelerated and traditional CPT. In this case, aggregate A2 was classed as alkali-reactive by the accelerated CPT and as not alkali-reactive by the traditional CPT.

The other three expansion limit criteria considered for the accelerated CPT (0.04% after 90, 120, or 150 days of testing) were found to be all suitable for a correct alkali-reactivity classification of the three aggregates tested: Aggregates A2 and A3 as not alkali-reactive and aggregate A1 as alkali-reactive.

In order to identify the best expansion limit criterion for the accelerated CPT, the TAL-evaluation approach was also considered in the elaboration of the test results of Figure 2 and Figure 3.

#### 4.3.2. TAL-Evaluation Approach

Considering the low changes in OH^−^ ion concentration of alkaline solutions during the accelerated CPT (Figure 1), and the low-alkali leaching from the concrete prisms subjected to the traditional CPT (Table 4), the initial alkali contents (initial Lac) of the concrete mixes investigated (Table 2) were used for the TAL evaluation of the three aggregates tested with both the traditional and accelerated CPT.

Figure 6 shows the TAL determination for the aggregates A1, A2 and A3 by using the expansion data obtained from the traditional CPT (Figure 3a–c). In this figure, the one year expansion of the concrete prisms made with a given aggregate is plotted as a function of the initial Lac. The horizontal line corresponding to the expansion limit of 0.04% is also drawn. The TAL value is determined as the abscissa of the intersection point between the expansion curve and the horizontal line.

TAL values of 3.9, 8.4, and 9.0 kg Na_2_Oeq/m^3^ were obtained from Figure 6 for aggregates A1, A2, and A3, respectively.

Based on these TAL values and according to the proposed classification of the alkali-reactivity of concrete aggregates [40] (TAL ≤ 2.8 kg Na_2_Oeq/m^3^: Highly alkali-reactive; 2.8 < TAL ≤ 5.5 kg Na_2_Oeq/m^3^: Moderately alkali-reactive; 5.5 < TAL ≤ 7.4 kg Na_2_Oeq/m^3^: Slowly alkali-reactive; TAL > 7.4 kg Na_2_Oeq/m^3^: Not alkali-reactive), aggregate A1 was classed as moderately alkali-reactive, aggregates A2 and A3 as not alkali-reactive.

Thus, the petrographic classification of aggregate A2 was also in contrast with the TAL evaluation of this aggregate.

In the case of the accelerated CPT, in order to identify the best expansion limit criterion, in particular the appropriate prism immersion time to be coupled with the expansion limit of 0.04%, the TAL values for the three aggregates were determined using the expansion data of Figure 2a–c corresponding to the immersion times of 90, 120, or 150 days.

Figure 7 shows the expansions of the concrete prisms made with a given aggregate measured at the considered immersion time, plotted as a function of the initial alkali content (initial Lac) of the concrete mixes investigated (Table 2). In this figure, the expansion limit of 0.04% is also reported as a dotted horizontal line.

The TAL values determined from Figure 7 for the three aggregates are compared in Table 5 with the corresponding TAL values obtained from the traditional CPT.

As evidenced by the results of Table 5, a good agreement between the TAL values determined with the two test methods was obtained only if an expansion limit criterion of 0.04% after 120 days of testing was used for the accelerated CPT.

With respect to the traditional CPT, the use of an expansion limit criterion of 0.04% after 90 days of testing significantly overestimated the TAL values of the three aggregates. Conversely, the TAL values were always underestimated using an expansion limit criterion of 0.04% after 150 days of testing, especially for aggregate A2 (6.0 kg Na_2_Oeq/m^3^ with accelerated CPT against 8.4 kg Na_2_Oeq/m^3^ with traditional CPT).

## 5. Conclusions

Based on the test results relevant to three natural aggregates of different alkali–silica reactivity, the following conclusions can be drawn.
(1)The concrete prism expansion test in alkaline (NaOH + KOH) solutions at 38 °C, proposed in this study as an accelerated CPT, appears to be a promising test method for assessing the alkali–silica reactivity of concrete aggregates. In this test, the concrete prisms are immersed in alkaline solutions (external solutions) simulating the pore liquid of the concrete mixes made with different alkali contents (initial Lac).(2)The compositions of external solutions can be designed by a simplified approach assuming total dissolution of cement alkalis and taking a ratio between the mass fractions of effective water consumed by cement hydration and of alkalis uptaken by cement hydrates (β_a_/β_w_ ratio) equal to unity. This approach is in an acceptable agreement with literature empirical equations correlating pore solution alkalinity of hardened Portland cement mixes with total alkali content of cement.(3)The changes in the OH^−^ ion concentration of the external solutions measured during the accelerated CPT are very low in virtue of the high ratio between the alkaline solution and the initial pore liquid volumes of concrete prisms adopted in this test (V_as_/V_pl_ = 10). As a result, there is a direct relationship between prism expansion and initial alkalinity of external solution and, hence, between expansion and initial Lac of concrete mix.(4)An expansion limit criterion of 0.04% after 120 days of testing is an appropriate judgement criterion for correlating the test results obtained from the proposed accelerated CPT and the traditional CPT (concrete prism expansion test at 38 °C and RH > 95%).(5)A reduction of the test duration from 365 to 120 days may be achieved using the proposed test method in place of the traditional CPT. This reduction of test duration would be of particular interest in the field of factory production control of concrete aggregates.

However, validation of the proposed accelerated CPT by a wide variety of concrete aggregates with different lithological/petrographic characteristics is needed prior to its possible use as a rapid and reliable test method for the alkali-reactivity assessment of concrete aggregates, through both pass-fail and TAL-evaluation approaches.

## Figures and Tables

**Figure 1 materials-13-00288-f001:**
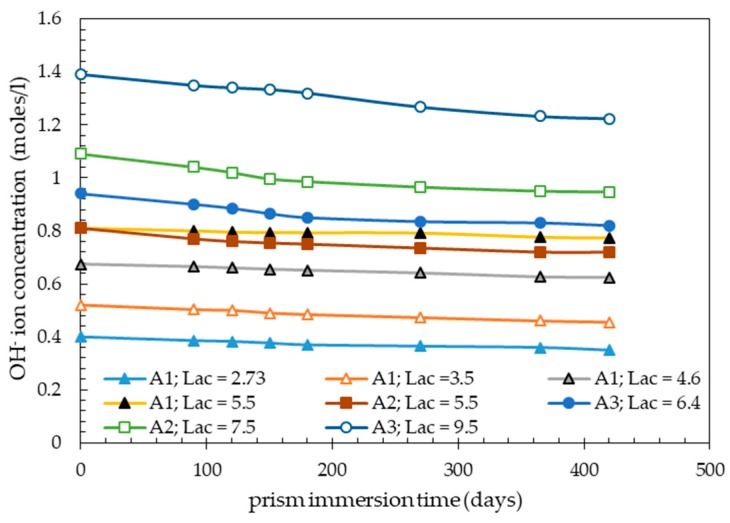
Changes in the OH^−^ ion concentration of alkaline solutions during the accelerated CPT.

**Figure 2 materials-13-00288-f002:**
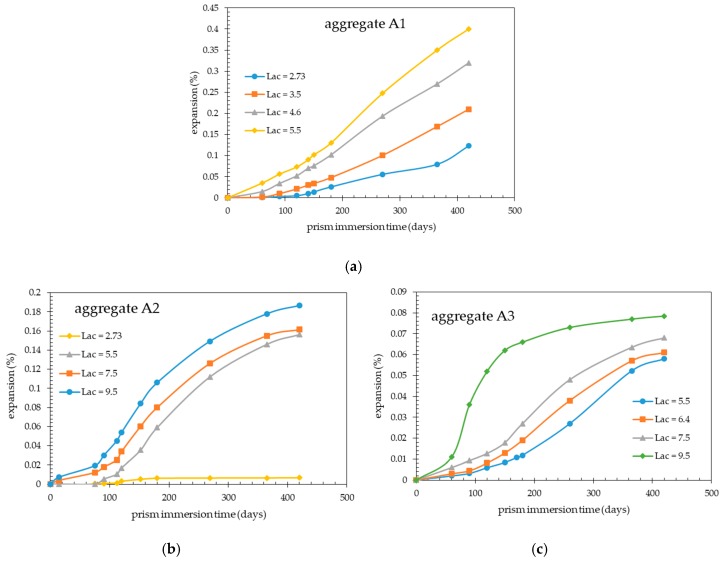
Expansions of concrete prisms with different aggregates and Lac values (kg Na_2_Oeq/m^3^) during the accelerated CPT: (**a**) Aggregate A1; (**b**) aggregate A2; (**c**) aggregate A3.

**Figure 3 materials-13-00288-f003:**
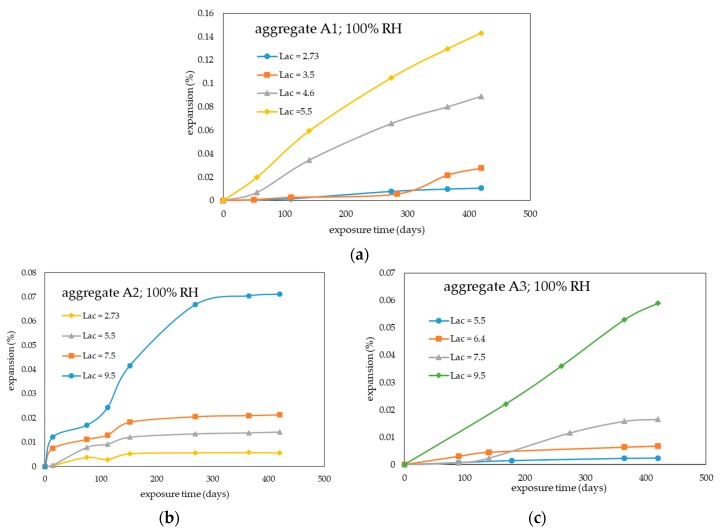
Expansions of concrete prisms with different aggregates and Lac values (kg Na_2_Oeq/m^3^) during the traditional CPT: (**a**) Aggregate A1; (**b**) aggregate A2; (**c**) aggregate A3.

**Figure 4 materials-13-00288-f004:**
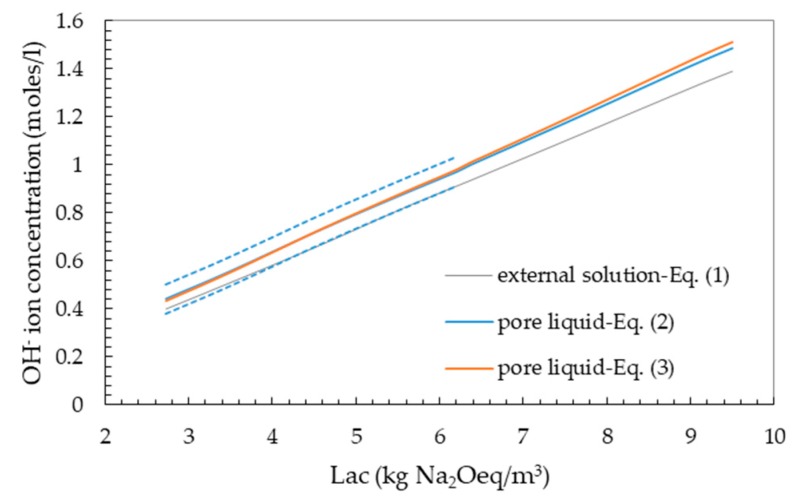
Comparison between the hydroxyl ion concentrations in the external solutions used in the immersion tests and those calculated for concrete pore solutions by using the empirical equations from [29] (Equation (2)) or [7] (Equation (3)).

**Figure 5 materials-13-00288-f005:**
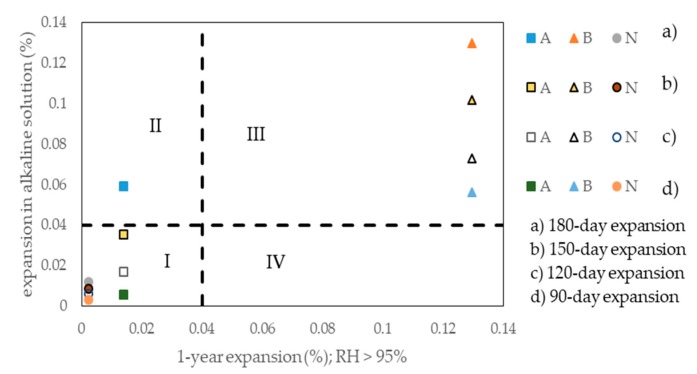
Comparison between the reactivity diagnoses for the three aggregates tested with the traditional CPT and the accelerated CPT using different expansion limit criteria.

**Figure 6 materials-13-00288-f006:**
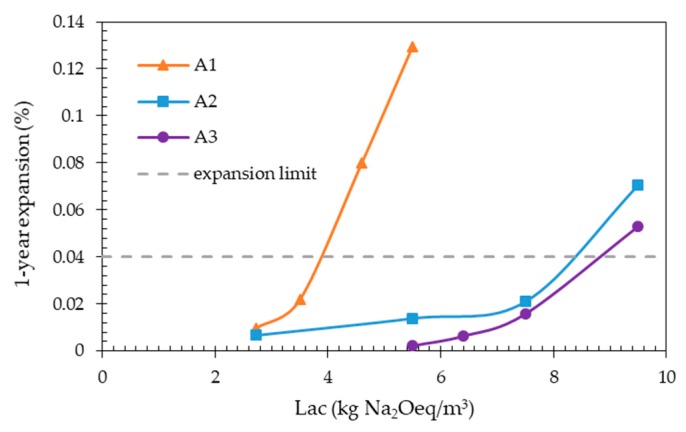
Determination of the threshold alkali level (TAL) values for the three aggregates tested with the traditional CPT.

**Figure 7 materials-13-00288-f007:**
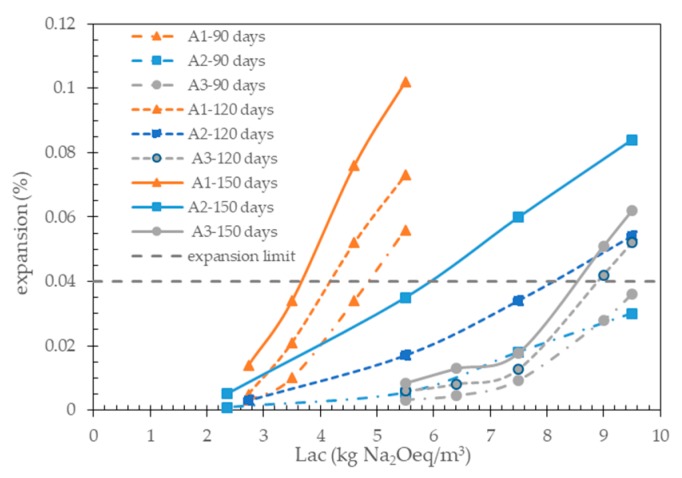
Determination of the TAL values for the aggregates tested with the accelerated CPT using different expansion limit criteria.

**Table 1 materials-13-00288-t001:** Chemical and mineralogical composition of Portland cement used.

Oxide	(%)	Oxide	(%)
CaO	62.19	Mn_2_O_3_	0.01
SiO_2_	21.46	P_2_O_5_	0.05
Al_2_O_3_	3.07	TiO_2_	0.08
Fe_2_O_3_	5.12	L.O.I.	2.12
SO_3_	2.62		
Na_2_O	0.34	Bogue composition	(%)
K_2_O	0.43	C_3_S	54.2
Na_2_Oeq ^1^	0.62	C_2_S	20.7
MgO	2.04	C_2_F	0.53
SrO	0.02	C_4_AF	14.6

^1^ % Na_2_Oeq = %Na_2_O + 0.658 %K_2_O.

**Table 2 materials-13-00288-t002:** Compositions of the concrete mixes investigated.

Component (kg/m^3^)								
Low-alkali CEM I 42.5 N		440	440	440	440	440	440	440
Aggregate (ssd) ^1^	A1	1670	1670	1670	1670	-	-	-
	A2	1660	-	-	1660	-	1660	1660
	A3	-	-	-	1730	1730	1730	1730
Effective water		220	220	220	220	220	220	220
NaOH (pellets)		-	0.99	2.41	3.58	4.74	6.16	8.74
Level of alkali content, Lac (kg Na_2_Oeq/m^3^)		2.73	3.5	4.6	5.5	6.4	7.5	9.5

^1^ Aggregate size gradation (% by mass): 0/4 mm = 40%; 4/12.5 mm = 35%; 12.5/22.5 mm = 25%.

**Table 3 materials-13-00288-t003:** Concrete alkali contents and corresponding chemical compositions of alkaline solutions used in the accelerated concrete prism test (accelerated CPT).

Initial Lac	NaOH	KOH	OH^−^
(kg Na_2_Oeq/m^3^)	(g/L)	(g/L)	(moles/L)
2.73	8.77	10.13	0.40
3.5	13.29	10.13	0.51
4.6	19.74	10.13	0.67
5.5	25.02	10.13	0.81
6.4	30.30	10.13	0.94
7.5	36.80	10.13	1.10
9.5	48.52	10.13	1.39

**Table 4 materials-13-00288-t004:** Alkali leaching from concrete prisms after one year of testing with the traditional CPT.

Concrete with Aggregate	Initial Lac	Na^+^ Leached	K^+^ Leached	Na_2_Oeq Leached	
	(kg Na_2_Oeq/m^3^)	(kg/m^3^)	(kg/m^3^)	(kg/m^3^)	(%)
A1	2.73	0.095	0.113	0.22	8.06
A1	3.5	0.145	0.113	0.29	8.29
A1	4.6	0.257	0.113	0.44	9.57
A1	5.5	0.348	0.110	0.56	10.18
A2	2.73	0.060	0.112	0.17	6.23
A2	5.5	0.220	0.105	0.38	6.91
A2	7.5	0.325	0.127	0.54	7.20
A2	9.5	0.485	0.118	0.75	7.89
A3	5.5	0.253	0.124	0.44	8.00
A3	6.4	0.348	0.123	0.57	8.91
A3	7.5	0.433	0.125	0.68	9.07
A3	9.5	0.710	0.122	1.05	11.05

**Table 5 materials-13-00288-t005:** Comparison between the TAL values of the three aggregates obtained from the traditional CPT and the accelerated CPT using different expansion limit criteria.

Aggregate	TAL Value (kg Na_2_Oeq/m^3^)			
	Traditional CPT	Accelerated CPT		
	0.04% after one Year	0.04% after 90 Days	0.04% after 120 Days	0.04% after 150 Days
A1	3.9	4.8	4.1	3.7
A2	8.4	10.7 *	8.1	6.0
A3	9.0	9.9 *	9.0	8.5

* Extrapolated value.

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
