# Peer review of "Assessment of Alkali–Silica Reactivity of Aggregates by Concrete Expansion Tests in Alkaline Solutions at 38 °C"

_materials, 2020, doi:10.3390/ma13020288_

Round 1
Reviewer 1 Report
The subject paper is interesting and its purpose complies with the journal’s aim and scope.
The manuscript gives a thorough overview of the current state of the art and presents all alternatives to CPT proposed to date, discussing their advantages and disadvantages. The methodology of the proposed accelerated concrete expansion test at 38oC is analytical and critically investigated. The results are adequately discussed.
I can detect no omissions or shortcomings in this manuscript and I can therefore suggest its acceptance as is.
Author Response
The subject paper is interesting and its purpose complies with the journal’s aim and scope.
The manuscript gives a thorough overview of the current state of the art and presents all alternatives to CPT proposed to date, discussing their advantages and disadvantages. The methodology of the proposed accelerated concrete expansion test at 38oC is analytical and critically investigated. The results are adequately discussed.
I can detect no omissions or shortcomings in this manuscript and I can therefore suggest its acceptance as is.
AA: The authors thank the reviewer for the positive evaluations expressed on the manuscript of which he shared the purpose in the need to have a test method suitable for the aggregate quality control supported by an adequate scientific background.
Reviewer 2 Report
The paper proposes a new methodology to assess alkali-reactivity of aggregate. Only three aggregates are tested, and a lot of supplementary work will be necessary to validate this method. Some parts are long and confuse and should be rewritten. At the end of the paper, it is not clear why this method improves expansion tests compared to usual tests.
L. 147-150: This text is not clear. Are the specimens put in high temperature only after 4 days of curing? It is very fast. Concrete is still very young at 4 days and hydration is still very active. Put concrete under high humidity and temperature can have important impact on the chemical composition of pore solution. It can not to be representative of real structures.
L. 160-165: In reality, the pore solution of concrete affected by ASR and immersed is never at the equilibrium. There are too many chemical reaction involved to have never leaching or enrichment: ASR leads to continuous modification of alkali concentration in the concrete pore solution so even if at the beginning of the test specimens and solution are real at chemical equilibrium, it will not be the case anymore after some days. As the volume of the solution is high compared to concrete porosity, such immersed tests lead always to enrichment.
L. 174-175: Why not the porosity? After cement setting, the volume of pore solution will be probably largely smaller than the volume of free water due to shrinkage.
L. 212-215: It depends on the nature of the aggregate. Some aggregates can lead to very fast expansion. Diffusion is not necessary the limiting factor but the reaction rate neither. In ASR, the two mechanisms are combined. Most of conclusions made without considering the combined mechanisms usually lead to incomplete conclusions.
L. 249-250: Why did you not use the same Lac for the three aggregate.
Figure 2: scattering of measurement should be added. Similar vertical scales could help to compare the expansions of the three aggregates.
L. 378: It seems very fast compared to relationships proposed for hardening of concrete in most standard.
L.379-380: It depends strongly of the aggregate nature and of the concrete composition.
Part 4.1: This part is very long and the objective is not clear. It should be rewritten.
Part 4.2.2: As external solution supplies large amount of alkali, TAL determined with this method can not to be representative of TAL in concrete in real conditions. TAL in real conditions should be probably larger. It should be discussed.
L. 408-410: Clarify.
L. 422: Why 0.04% was taken as a criterion in standards?
Author Response
The paper proposes a new methodology to assess alkali-reactivity of aggregate. Only three aggregates are tested, and a lot of supplementary work will be necessary to validate this method. Some parts are long and confuse and should be rewritten. At the end of the paper, it is not clear why this method improves expansion tests compared to usual tests.
AA: The authors highlighted in the original manuscript that the robustness of the proposed method will have to be ascertained by extending the experimentation to other aggregates. However the results of the work are promising and induce to invest further resources for the definitive validation of the method. In the
introduction, the various standardized or proposed methods to evaluate the reactivity of the aggregates are commented. In particular, it was pointed out that the CPT method at 38° C, standardized in many Countries and often used as a reference method, is not suitable for quality control in the production of aggregates due to the excessive length of test time (365 d). The proposed method may constitute an advantageous
alternative in the quality control for the significantly reduced test duration (120 d).This item has been included in the paragraph “Conclusions” of the revised manuscript.
L. 147-150: This text is not clear. Are the specimens put in high temperature only after 4 days of curing? It is very fast. Concrete is still very young at 4 days and hydration is still very active. Put concrete under high humidity and temperature can have important impact on the chemical composition of pore solution. It can not to be representative of real structures.
AA: The 38 °C test temperature cannot be considered particularly "high". This temperature does not destabilize the hydrated phases which are formed by hydration of the cement at 20 °C, including ettringite which is unstable quite over 40°C. It should be noted that temperatures of this level can be reached by
concrete structures under certain conditions of exposure or due to the thermal rise caused by the heat of hydration of the cement. It should also be considered that all laboratory test methods operate with standardized conditions in terms of concrete composition, alkali content, temperature and humidity in order to characterize the reactivity of the aggregates and not to verify the expansive behaviour of the relevant concretes used in the real structures.
L. 160-165: In reality, the pore solution of concrete affected by ASR and immersed is never at the equilibrium. There are too many chemical reaction involved to have never leaching or enrichment: ASR leads to continuous modification of alkali concentration in the concrete pore solution so even if at the beginning of the test specimens and solution are real at chemical equilibrium, it will not be the case
anymore after some days. As the volume of the solution is high compared to concrete porosity, such immersed tests lead always to enrichment.
AA: The authors agree with the comment about the lack of leaching in the case of immersion tests. Therefore, lines 160-164 have been deleted in the revised version.
L. 174-175: Why not the porosity? After cement setting, the volume of pore solution will be probably largely smaller than the volume of free water due to shrinkage.
AA: The determination of the porosity of the concrete after pre-curing would constitute a considerable experimental burden for a method addressed above all to the quality control and usable by the factory technological laboratories. For this reason, a simplified approach for designing the concentration of the immersion solution has been adopted. Anyway, the consistency of the used hydroxyl ion concentrations with the hydroxyl’s concentrations in the pore solution was assessed by a comparison with literature data
relevant the composition of pore solution of mature mortars and concretes. (see par. 4.1).
L. 212-215: It depends on the nature of the aggregate. Some aggregates can lead to very fast expansion. Diffusion is not necessary the limiting factor but the reaction rate neither. In ASR, the two mechanisms are combined. Most of conclusions made without considering the combined mechanisms usually lead to incomplete conclusions.
AA: The authors agree that the kinetics of the expansive phenomena deriving from the ASR can vary considerably depending on the nature of the aggregate; the expansion data recorded on the concretes of tested aggregates confirm this finding. Furthermore, they agree that chemical reaction and diffusion are
concomitant mechanisms, but they are nevertheless of the opinion that the rate of the chemical reaction is the limiting factor and that the ion diffusion through the solution can continually restore the alkali consumed by ASR in the pore liquid.
L. 249-250: Why did you not use the same Lac for the three aggregate.
AA: The adopted Lac values were selected according to the reactivity of aggregates resulting from the petrographic analysis and their behaviour in the field. So, higher Lac values were set for concretes containing aggregate A3 surely non-reactive.
Figure 2: scattering of measurement should be added. Similar vertical scales could help to compare the expansions of the three aggregates.
AA: The scattering of the expansion measurements is given in terms of standard deviation or variation coefficient of the triplicate specimens in the paragraph 2.2 – lines 184-186 in the original manuscript. The authors believe that it is better to maintain different scales of the figures in order to highlight the very different aggregate reactivity.
L. 378: It seems very fast compared to relationships proposed for hardening of concrete in most concrete standards.
AA: The statement reported in the original manuscript does not derive from author’s findings but it is supported by references 35 and 36 (37 and 38 in the revised manuscript). In the revised manuscript, this statement has been modified by replacing “exhibit” with “may exhibit” (line 396). If it is considered the temperature values of test methods used for ASR assessing, the data reported in the above references are not in substantial contrast with the figures deduced from the Eurocode model for strength development vs
age of concrete [Eurocode 2: Design of concrete structures EN 1992-1-1].
L.379-380: It depends strongly of the aggregate nature and of the concrete composition.
AA: The authors have considered the reviewer’s comments and they have accordingly revised the manuscript, Lines 397-400
Part 4.1: This part is very long and the objective is not clear. It should be rewritten.
AA: The authors have considered the reviewer’s comments and they have accordingly revised the manuscript by dividing paragraph 4.1 in two parts having two different scopes (see pages 11-14) and by reorganizing the text of the original manuscript:
1) Par. 4.1 (new): Verification of the consistency of the experimental approach in the accelerated CPT development.
2) Par. 4.2 (new): interpretation of the results presented in paragraph 4.1, based on the factors affecting the pore liquid alkalinity.
Part 4.2.2: As external solution supplies large amount of alkali, TAL determined with this method can not to be representative of TAL in concrete in real conditions. TAL in real conditions should be probably larger. It
should be discussed.
AA: The application of the proposed accelerated method for the determination of the TAL parameter was made with the sole purpose of defining appropriate expansion limits, taking as reference values those obtained with the traditional CPT. The extension of the method to the determination of the TAL must be the subject of future investigations. It should be noted, however, that the TAL values obtained with the
traditional CPT method are relevant to standardized test conditions and therefore their use as a criterion for formulating the mix design of real concrete for the prevention of ASR must be evaluated case by case.
L. 408-410: Clarify.
AA: The authors have considered the reviewer’s comments and they have accordingly revised the manuscript at Lines 262-269
L. 422: Why 0.04% was taken as a criterion in standards?
AA: An expansion limit of 0.04% is specified in ASTM, CSA, UNI standards.
Reviewer 3 Report
P1 Line 34: So far it is only with sodium or potassium. Lithium is rather on the opposite side. So, better to say: hydroxyl ions in concrete pore solution associated with sodium and potassium.
P1 Line 36: combine this single sentence paragraph with the previous paragraph.
P1 Line 39: Replace "there is... proposed" by "many test methods are available in the literature"
P2 Line 42: provide citations for the mortar bar and cp tests.
L2P49: replace "diffuse" with "used"
P2L67: A different response of the increase in temperature has been explained in: Gautam, B. P., and Panesar, D. K. (2017). “The effect of elevated conditioning temperature
on the ASR expansion, cracking and properties of reactive Spratt aggregate concrete.”
Construction and Building Materials, 140, 310–320. Please review the paper and update the paragraph.
Last paragraph of the Introduction section: Change it to the present tense.
P3L101: Delete the redundant term "tested"
In P3 you say A2 and A3 as non-reactive but in the first two paragraphs in P4, you describe A2 as reactive. It is confusing; pls correct it. Do you mean to contrast the observation on field and on petrographic analysis? If so, what is your stake about A2? Is it reactive or not? Don't pass your confusion to the readers!
P3L109: Designation of A1, A2, A3 is fine but why don't you disclose their proper name and source? The aggregates shall be introduced by their name, source and mention to literature
describing them, if available.
P4L133 and Table 2: Make it clear what "Lac" stands for.
P4 Last paragraph: you mention a range of Lac and also say that the values corresponded to the initial content. An initial content can't be a range?
P5L148: Here you use >95%, previously you say 100%. Please be precise. >95% is the accurate one as 100% may not be ensured always.
P5L155: why "no wick"?
P5L159: Please use succinct word. Replace "resulted to be" by "were"
The same applies to other places where you use the expletives such as in P5L160: "It is of fundamental importance that..."
P5L165 and the paragraph: It appears that you implicitly assume a constant pore solution concentration over time. This is unlikely. Please elaborate the assumption/limitation.
P5L174: Free water is not the appropriate term. It is the mixing water.
P6L196: Did you vary your alkali solution to reflect this change in pore ion concentration?
With diffusion, the concentration of the external solution is likely to reduce. Did you add alkalies to make up the concentration?
P6L214: But the concentration shall be decreasing over time rather than assuming a constant.
P7L237: Not clear how you got the number 10. Your initial pore liquid volume is 220*.07*.07*.26=0.65 l and you submerged in 2.8 l solution?? And what is the basis to claim the attenuation with the ratio of 10?
P8L257: delete "both", replace "and" by "or".
P8L259: What is the rationality behind stopping the test at 420 days?
P8L266: Further elaborate about Fig 2b rather than just writing one sentence for one curve.
P9L286: Replace "exception made" by "except".
P10L293: It is almost meaningless to state a range as 1.5 to 20. Better you specify the ratio after some test duration or illustrate it through a table or a plot of the ratio.
P10L310: For the same alkali content, highest leaching for the most reactive aggregate is COUNTERINTUITIVE. Please explain.
P11L316: Make it clearer by associating this point with the reduced size of the container.
Not clear how Eq. 6 was derived from Eq 5? What about the numbers?
P13L404: But it is only valid if the alkalies in the water consumed during hydration are embedded in the gel. No convincing argument whether the alkali ions will be left out or associated.
P13L424: Not sure of UNI but 5.25 as per ASTM.
Overall comment: Standard methods exist for extracting the pore solution of concrete and measuring its ion analysis. Without even analyzing pore solution, the claims about the ion concentration inside and outside the concrete appear unsupported and remain theoretical only.
Not even a literature is reported regarding the pore solution extraction and analysis. This issue must be addressed before publishing this paper.
Author Response
P1 Line 34: So far it is only with sodium or potassium. Lithium is rather on the opposite side. So, better to
say: hydroxyl ions in concrete pore solution associated with sodium and potassium.
AA: The authors have considered the reviewer’s comment and they have accordingly revised the manuscript,
Line 34
P1 Line 36: combine this single sentence paragraph with the previous paragraph.
AA: Done, Line 36.
P1 Line 39: Replace "there is... proposed" by "many test methods are available in the literature"
AA: The authors have considered the reviewer’s comment and they have accordingly revised the manuscript,
Line 40
P2 Line 42: provide citations for the mortar bar and cp tests.
AA: The authors have considered the reviewer’s comment and they have added previously cited references,
Line 44.
P2 L49: replace "diffuse" with "used"
AA: Done, Line 50.
P2 L67: A different response of the increase in temperature has been explained in: Gautam, B. P., and
Panesar, D. K. (2017). “The effect of elevated conditioning temperature on the ASR expansion, cracking and
properties of reactive Spratt aggregate concrete. ”Construction and Building Materials, 140, 310–320.
Please review the paper and update the paragraph.
AA: The authors have considered the reviewer’s comment and they have included in the revised manuscript
the above mentioned literature references, Lines 75-76.
Last paragraph of the Introduction section: Change it to the present tense.
AA: Done, Lines 97 and 101.
P3 L101: Delete the redundant term "tested"
AA: Done, Line 103.
In P3 you say A2 and A3 as non-reactive but in the first two paragraphs in P4, you describe A2 as reactive. It
is confusing; pls correct it. Do you mean to contrast the observation on field and on petrographic analysis?
If so, what is your stake about A2? Is it reactive or not? Don't pass your confusion to the readers!
AA: The authors have considered the reviewer’s comment and they have accordingly revised the manuscript,
Lines 127-129.
P3 L109: Designation of A1, A2, A3 is fine but why don't you disclose their proper name and source? The
aggregates shall be introduced by their name, source and mention to literature describing them, if
available.
AA: The aggregates used are commercial products and an agreement was signed with the providers
according to which their origin cannot be revealed in order to avoid market problems.
P4 L133 and Table 2: Make it clear what "Lac" stands for.
AA: The authors have considered the reviewer’s comment and they have accordingly revised the manuscript,
Line 134 and Table 2
P4 Last paragraph: you mention a range of Lac and also say that the values corresponded to the initial
content. An initial content can't be a range?
AA: The authors have considered the reviewer’s comment and they have accordingly revised the manuscript,
Lines 136-137.
P5 L148: Here you use >95%, previously you say 100%. Please be precise. >95% is the accurate one as 100%
may not be ensured always.
AA: The authors have considered the reviewer’s comment and they have modified the manuscript
accordingly.
P5 L155: why "no wick"?
AA: In accordance with the UNI 11604 standard, wick was not placed in the test containers.
P5 L159: Please use succinct word. Replace "resulted to be" by "were"
The same applies to other places where you use the expletives such as in P5L160: "It is of fundamental
importance that..."
AA: The authors have considered the reviewer’s comment and they have modified the manuscript
accordingly.
P5 L165 and the paragraph: It appears that you implicitly assume a constant pore solution concentration
over time. This is unlikely. Please elaborate the assumption/limitation.
AA: The assumption was just elaborated and clarified on page 7/8 - lines 242-249 of the original manuscript.
P5 L174: Free water is not the appropriate term. It is the mixing water.
AA: The authors prefer to replace in the revised manuscript, the term “free water” with “effective water” in
line with the European concrete standard EN 206. This modification was applied throughout the text and to
Table 2.
P6 L196: Did you vary your alkali solution to reflect this change in pore ion concentration? With diffusion,
the concentration of the external solution is likely to reduce. Did you add alkalis to make up the
concentration?
AA: The monitoring of the alkalinity of the external solution has shown a very little OH-
concentration
decrease during the test time, so it was not considered necessary to add alkalis to the external solution
during the test. Furthermore, it was necessary to verify the suitability of the test apparatus in attenuating
the OH-
ion concentration changes in the external solutions during the accelerated CPT.
P6 L214: But the concentration shall be decreasing over time rather than assuming a constant.
AA: This item is discussed and clarified in paragraph 4.2 of the revised manuscript.
P7 L237: Not clear how you got the number 10. Your initial pore liquid volume is 220*.07*.07*.26=0.65 l
and you submerged in 2.8 l solution?? And what is the basis to claim the attenuation with the ratio of 10?
AA: The authors checked the value of the ratio between the volume of the external solution (2.8 l) and that
of the pore solution present in the concrete prisms (0.28028 l). The ratio results 9.99 so ratio of 10 is
confirmed .They point out that the product 220x.07x.07x.26 gives 0.28028 l not 0.65 l as the reviewer has
written.
P8 L257: delete "both", replace "and" by "or".
AA: Done, Line 249
P8 L259: What is the rationality behind stopping the test at 420 days?
AA: To better appreciate the long-term expansion trend and to verify the reliability of the expansion values
at the testing time of 365 days, established by UNI or ASTM standards.
P8 L266: Further elaborate about Fig 2b rather than just writing one sentence for one curve.
AA: The data in Figure 2(b) along with those of Figures 2(a) and 2(c) have been largely used for obtaining
the curves reported in Figure 7. However, a further comment on Figure 2(b) has been added at Lines 262 -
269.
P9 L286: Replace "exception made" by "except".
AA: Done, Line 289.
P10 L293: It is almost meaningless to state a range as 1.5 to 20. Better you specify the ratio after some test
duration or illustrate it through a table or a plot of the ratio.
AA: The authors have considered the reviewer’s comment and consequently they have included at the lines
296-298 of the revised manuscript the ratios of the ultimate expansions determined with the two test
methods on each aggregate for the concrete mix with Lac=5.5 kg Na2Oeq/m3
. (A1=2.8; A2=11; A3=24)
P10 L310: For the same alkali content, highest leaching for the most reactive aggregate is
COUNTERINTUITIVE. Please explain.
AA: The explanation is provided in the revised text, lines 314-315
P11 L316: Make it clearer by associating this point with the reduced size of the container.
AA: Done, Line 321.
Not clear how Eq. 6 was derived from Eq 5? What about the numbers?
AA: The reworking of paragraph 4.1 made it possible to reduce the number of equations and simplify the
processing; therefore equation (6) has now been renumbered as equation (5).
P13 L404: But it is only valid if the alkalis in the water consumed during hydration are embedded in the gel.
No convincing argument whether the alkali ions will be left out or associated.
AA: The authors have considered the reviewer’s comment in reworking the new paragraph 4.2.
P13 L424: Not sure of UNI but 5.25 as per ASTM.
AA: 5.5 kg Na2Oeq/m3
corresponds to the Lac value specified by the two methods object of bibliographic
references [6] and [12], corresponding to RILEM AAR3 and UNI 11604, respectively. The text has been
revised accordingly (Line 452).
Overall comment: Standard methods exist for extracting the pore solution of concrete and measuring its
ion analysis. Without even analyzing pore solution, the claims about the ion concentration inside and
outside the concrete appear unsupported and remain theoretical only. Not even a literature is reported
regarding the pore solution extraction and analysis. This issue must be addressed before publishing this
paper.
As more clearly described in paragraph 4.1 of the revised manuscript, the OH- ion concentrations of the
external alkaline solutions were compared to those calculated for the concrete pore liquid by using two wellknown empirical equations available in the literature. The objective was to verify the suitability of the
experimental approach adopted in the development of the proposed accelerated CPT. The two empirical
equations developed by Helmuth [29] and by Thomas et al. [7] were obtained by using the results of many
studies dealing with the expression of pore solution from cementitious mixes. The citations of these studies
are located in the papers of Helmuth [29] and Thomas et al. [7].
As fas as the lack of our experimental results of pore liquid expression, it must be considered that the
primary objective of our study was to develop a rapid and reliable test method to be applied especially in
the field of factory production control of concrete aggregates. Therefore, comparison between the proposed
test method and the traditional CPT was mostly aimed at identifying the best expansion limit criterion for
the accelerated CPT. The promising results obtained in this study encourages the prosecution of the research
and, in that case, expression of pore liquid from concrete prisms subjected to expansion tests will have to be
carried out. This concept has also been included in the revised manuscript, at page 14, Lines 436-439.
Round 2
Reviewer 2 Report
The modifications have been performed.